# Entropy-Triggered Retraining as Nonequilibrium Entropy Production in Deployed Machine Learning Systems

## Abstract

Machine learning models deployed in nonstationary environments inevitably experience performance degradation due to data drift. While numerous drift detection heuristics exist, most lack a dynamical interpretation and provide limited guidance on how retraining decisions should be balanced against operational cost. In this work, we propose an entropy-based retraining framework grounded in nonequilibrium statistical physics. Interpreting drift as probability flow governed by a Fokker–Planck equation, we quantify model–data mismatch using relative entropy and show that its time derivative admits an entropy-balance decomposition featuring a nonnegative entropy production term driven by probability currents. Guided by this theory, we implement an entropy-triggered retraining policy using an exponentially weighted moving-average (EWMA) control statistic applied to a streaming kernel density estimator of the Kullback–Leibler divergence. We evaluate this approach across multiple nonstationary data streams. In synthetic, financial, and web-traffic domains, entropy-based retraining achieves predictive performance comparable to frequent retraining while reducing retraining frequency by one to two orders of magnitude. However, in a challenging biomedical ECG setting, the entropy-based trigger underperforms the maximum-frequency baseline, highlighting limitations of feature-space entropy monitoring under complex label-conditional drift.

Mathematical machine learning, Fokker–Planck equations, relative entropy, Lyapunov functionals, data drift, stochastic processes, nonequilibrium dynamics [1]

## 1 Introduction

Machine learning models deployed in real-world environments operate under the implicit assumption that training and deployment data are drawn from the same distribution. In practice, this assumption is routinely violated. Changes in population behavior, sensing mechanisms, feedback effects, or external conditions induce data drift, leading to gradual or abrupt degradation in predictive performance across applications ranging from finance and healthcare to recommender systems and autonomous decision-making.

Despite its prevalence, data drift is often treated primarily as a detection problem rather than a decision problem. Common monitoring strategies, such as heuristic distributional tests, confidence-based alarms, or delayed accuracy feedback, typically answer the question of *whether* a change has occurred, but provide limited guidance on *when* retraining is warranted and how retraining frequency should be balanced against operational cost. In deployed systems, retraining incurs nontrivial expense in computation, data labeling, validation, and organizational overhead. As a result, overly sensitive triggers lead to excessive retraining, while conservative policies permit degradation to accumulate.

This paper takes a different point of view. We treat deployment as an evolving stochastic system whose feature distribution changes in time. From this perspective, the central object is not a particular alarm statistic, but the *dynamics* by which the deployment-time data distribution drifts away from the distribution

---

[1]AI-based tools (ChatGPT, OpenAI) were used in a limited assistive role for editing, LaTeX formatting, symbolic checks, and code drafting; all scientific content and conclusions are the author's own.

implicitly encoded by a fixed deployed model. Nonequilibrium statistical physics provides a principled framework for describing irreversible processes through entropy production and probability flow: stochastic dynamics generate probability currents that drive systems away from reference states, producing entropy in the process. Central to this theory is the Fokker–Planck equation, which governs the evolution of probability densities and admits entropy-based functionals that quantify irreversibility.

The modern theory of irreversible processes and entropy production originates in the foundational work of Onsager, which established reciprocity relations and identified entropy production as a structural feature of nonequilibrium systems Onsager (1931). Subsequent developments formalized the role of probability currents in driving irreversibility and entropy generation in stochastic systems Schnakenberg (1976). Within this framework, entropy production is not an incidental artifact of noise, but a consequence of persistent probability flow away from equilibrium.

We leverage this framework to reinterpret model degradation under drift as an entropy-producing dynamical process. Specifically, we model the deployment-time data distribution as a probability flow and quantify model–data mismatch using relative entropy. This yields a label-free signal that is tied directly to the dynamics of drift. Retraining then becomes an intervention that resets the reference distribution associated with the deployed model, reducing accumulated mismatch at an operational cost. This viewpoint motivates entropy-triggered retraining policies that balance predictive performance against retraining frequency by responding to the underlying probability flow rather than to delayed label-dependent performance collapse.

**Contributions** The contributions of this work are threefold:

- We provide a dynamical interpretation of model degradation under data drift as nonequilibrium entropy production governed by Fokker–Planck probability flow.

- We establish a relative-entropy mismatch functional whose evolution is driven by probability currents, motivating entropy-triggered retraining as a principled, label-free intervention strategy.

- We empirically evaluate entropy-triggered retraining with an EWMA-based alarm on a streaming KL estimator, and quantify the resulting cost–performance tradeoffs across multiple drifting domains.

**Organization** Section 2 introduces a probability-flow model of drift. Section 3 defines mismatch entropy and derives its evolution along the flow. Section 4 interprets retraining as an external control and specifies the EWMA trigger used in experiments. Section 5 evaluates entropy-triggered retraining empirically. Appendix A provides the entropy-balance derivation, and Appendix B specifies the EWMA thresholding mechanism.

## 2 Drift as Probability Flow

Let $X_t \in \mathbb{R}^d$ denote the feature vector observed at deployment time $t$, where $t$ indexes a discrete-time data stream. In deployed systems, the underlying mechanisms generating drift are typically unknown and need not admit an explicit parametric description. Nevertheless, it is often useful to interpret distributional change through the lens of continuous-time stochastic dynamics.

As a conceptual model, we consider a stochastic process whose evolution is described by the stochastic differential equation

$$dX_t = a(X_t, t)\, dt + B(X_t, t)\, dW_t, \tag{1}$$

where $a$ is a drift field, $B$ a diffusion matrix, and $W_t$ standard Brownian motion. This formulation is not assumed to govern the data-generating process in our experiments; rather, it serves as an interpretive framework for understanding how probability distributions evolve under persistent, nonequilibrium forcing.

Under standard regularity assumptions, the probability density $p(x, t)$ associated with such a process satisfies the Fokker–Planck equation

$$\partial_t p(x, t) = -\nabla \cdot J(x, t), \tag{2}$$

where the probability current is given by

$$J(x,t) = a(x,t)p(x,t) - \nabla \cdot (D(x,t)p(x,t)), \quad D = \tfrac{1}{2}BB^\top. \tag{3}$$

The Fokker–Planck equation provides a canonical description of probability flow and admits a rich analytical structure; see Risken (1989) for a comprehensive treatment.

In deployed machine learning systems, neither the drift field $a$ nor the diffusion tensor $D$ is observed directly. Instead, the system is monitored through finite samples drawn from the time-varying feature distribution. From this perspective, the probability current $J(x,t)$ is not an observable object, but a conceptual quantity encoding the irreversible flow of mass in feature space induced by drift.

Nonzero probability currents correspond to nonequilibrium dynamics and provide a mechanism by which a fixed deployed model becomes misaligned with its environment. In nonequilibrium statistical physics, such currents are the fundamental drivers of irreversibility and entropy production, encoding sustained deviation from equilibrium through persistent probability flow Schnakenberg (1976). This viewpoint motivates the use of entropy-based functionals as monitoring signals for deployment-time mismatch, even when only discrete-time empirical observations are available.

## 3 Mismatch Entropy and Its Dynamical Interpretation

We quantify model–data mismatch using the relative entropy

$$D(t) = D_{\mathrm{KL}}\big(p(\cdot,t) \,\|\, q_{\mathrm{ref}}\big), \tag{4}$$

which measures the statistical divergence between the evolving deployment-time data distribution $p(x,t)$ and a fixed reference distribution $q_{\mathrm{ref}}$ associated with the deployed model (e.g., the training distribution or an updated reference after retraining). From an information-theoretic perspective, relative entropy provides a natural and asymmetric measure of distributional mismatch that is independent of labels or model outputs Cover & Thomas (2006).

Differentiating $D(t)$ along the Fokker–Planck probability flow yields

$$\frac{d}{dt}D(t) = \int_{\mathbb{R}^d} J(x,t) \cdot \nabla \log \frac{p(x,t)}{q_{\mathrm{ref}}(x)} \, dx, \tag{5}$$

where boundary terms are assumed to vanish. This identity makes explicit that the temporal evolution of mismatch is driven by probability currents in feature space induced by drift.

More generally, along nonequilibrium probability flows the entropy rate admits a decomposition analogous to entropy-balance relations in stochastic thermodynamics, featuring a nonnegative entropy production term driven by probability currents and a residual "housekeeping" term that captures sustained external driving Seifert (2005). In the present work, this decomposition serves as a theoretical motivation rather than a directly estimated quantity. The full derivation is provided in Appendix A.

**Remark 1 (Lyapunov and nonequilibrium interpretation)** *When $q_{\mathrm{ref}}$ coincides with an appropriate stationary reference for the flow, the mismatch entropy $D(t)$ behaves as a Lyapunov functional and decays monotonically. Under driven nonequilibrium drift, $D(t)$ need not be monotone; nevertheless, its evolution is constrained by an entropy-balance relation with a nonnegative entropy production term. Increases in mismatch reflect the effect of sustained probability currents relative to the fixed deployed reference distribution.*

This distinction matters operationally. Loss-based retraining triggers are necessarily reactive because they depend on labels and respond only after predictive performance has already degraded. In contrast, mismatch entropy is a property of the deployment-time feature distribution and can provide an early warning signal in settings where feature-space drift correlates with predictive risk (e.g., under covariate shift), indicating when retraining may become necessary before substantial performance degradation is observed.

However, it is important to note a fundamental limitation of this approach. The mismatch entropy $D(t)$ monitors changes in the marginal feature distribution $p(x)$ and is therefore most sensitive to covariate drift.

In settings where performance degradation is driven primarily by changes in the conditional distribution $p(y \mid x)$ with relatively stable $p(x)$, feature-space entropy may fail to provide a timely or sufficient signal for retraining. This limitation is examined empirically in Section 5, where entropy-triggered retraining underperforms in a biomedical ECG domain characterized by strong label-conditional variability.

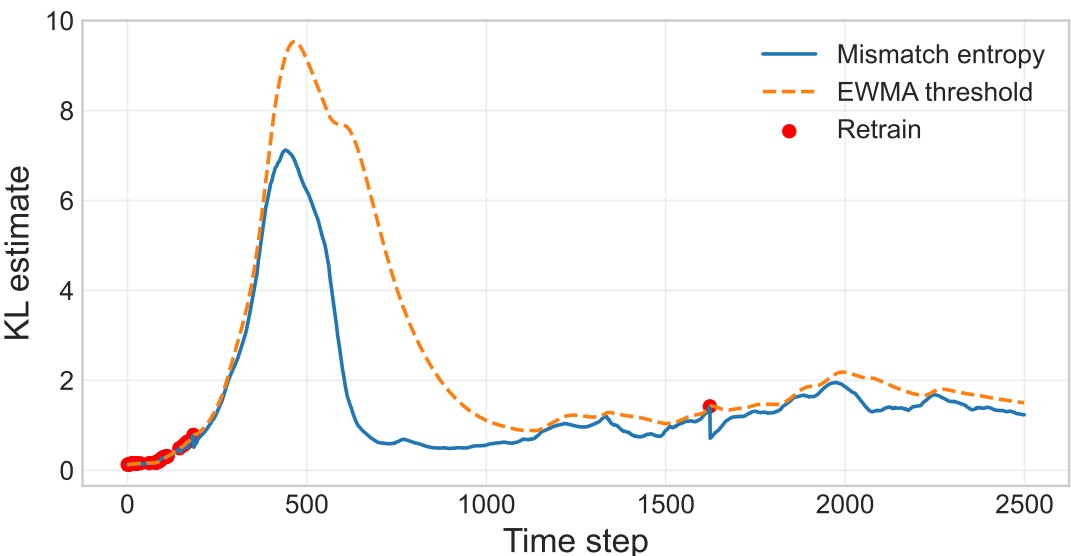

Figure 1: Evolution of an estimated mismatch entropy under drift. The threshold defines entropy-triggered retraining events. Small negative values may arise from finite-sample KDE estimation error; the true relative entropy is nonnegative.

**From mismatch to intervention**   If the deployment-time distribution evolves under persistent probability currents, then mismatch relative to a fixed deployed reference will generally change over time, and sustained drift corresponds to nonequilibrium entropy production in the underlying dynamical system. Maintaining predictive performance therefore requires an intervention that updates the deployed reference. In deployed learning systems, this intervention takes the form of retraining.

## 4   Retraining as External Work and EWMA Triggers

Retraining corresponds to resetting the reference distribution $q_{\mathrm{ref}}$ by incorporating newly observed data, thereby reducing accumulated mismatch entropy at the cost of computation, data acquisition or labeling, validation, and operational disruption. This mirrors the role of external work in nonequilibrium statistical physics, where entropy reduction requires resources supplied by an external agent.

**Practical KL estimator (as implemented)**   In experiments, the mismatch entropy is estimated online using kernel density estimates (KDEs). At time index $t$ (discrete time), let $\{x_i\}_{i=1}^n$ denote a recent "fit" batch. We fit a KDE $\hat{p}_t$ on the fit batch and maintain a reference KDE $\hat{q}_{\mathrm{ref}}$ (updated only when an entropy-triggered retraining occurs). The empirical KL estimator used is

$$\widehat{D}_t = \frac{1}{n} \sum_{i=1}^n \Big( \log \hat{p}_t(x_i) - \log \hat{q}_{\mathrm{ref}}(x_i) \Big). \tag{6}$$

**EWMA thresholding**   Instead of using a fixed sliding-window quantile threshold, retraining events are triggered using an exponentially weighted moving average (EWMA) alarm on the monitored statistic. Let $s_t$

denote a scalar monitoring signal (either $\widehat{D}_t$ for entropy-triggered retraining, or an online log-loss estimate for performance-triggered retraining). We maintain an EWMA mean $\mu_t$ and EWMA variance $v_t$:

$$\mu_t = (1 - \alpha)\mu_{t-1} + \alpha s_t, \tag{7}$$

$$v_t = (1 - \alpha)v_{t-1} + \alpha(s_t - \mu_t)^2. \tag{8}$$

The smoothing coefficient is parameterized by a half-life $h$ via

$$\alpha = 1 - 2^{-1/h}. \tag{9}$$

A retraining event is triggered when the standardized deviation exceeds a fixed level $k$:

$$z_t = \frac{s_t - \mu_t}{\sqrt{v_t} + \varepsilon} > k, \tag{10}$$

with $\varepsilon > 0$ a numerical stabilizer. In the experiments reported here, we use $h = 50$ and $k = 2.0$ for both entropy- and performance-based triggers, matching the experimental script.

## 5 Experimental Evaluation

### 5.1 Setup

We compare four retraining strategies implemented in a unified streaming evaluation pipeline:

- **No retraining:** train once on an initial window and never update.

- **Daily retraining (maximum-frequency baseline):** retrain at every time step on a rolling training window.

- **Entropy-triggered retraining:** compute the KL estimator $\widehat{D}_t$ via KDE and trigger retraining using the EWMA rule in Section 4; on trigger, update both the classifier and the KDE reference.

- **Performance-triggered retraining:** compute an online log-loss estimate and trigger retraining using the same EWMA rule; on trigger, update the classifier (labels assumed available).

Across all domains, the classifier is logistic regression trained on z-scored features. Feature normalization is fixed using the mean and standard deviation of the initial training window and applied to all subsequent data, including after retraining. Predictive performance is evaluated online using Bernoulli log loss, averaged over a small evaluation batch at each time step.

We evaluate these strategies on four drifting domains:

- **Synthetic Gaussian drift:** two-dimensional Gaussian features with gradually shifting mean and variance, and a rotating logistic decision boundary. Each time step corresponds to a full block of samples.

- **Finance (SPY):** daily market features (returns, rolling volatility, RSI, momentum) used to predict the next-day return sign over a long historical window.

- **Web traffic (Wikipedia pageviews):** daily pageviews for a popular article (Bitcoin), with log-transformed volume, rolling statistics, momentum, and weekly seasonality features.

- **Biomedical (MIT-BIH ECG):** beat-level features extracted from ECG segments, labeled as normal versus abnormal beats using MIT-BIH annotations.

### 5.2 Results

Table 1 reports a cross-domain summary of average log loss and retraining frequency. Across the synthetic, financial, and web-traffic domains, entropy-triggered retraining achieves predictive performance close to the daily retraining baseline while requiring only a small fraction of the retraining events. In these settings, entropy monitoring successfully captures deployment-time drift that is relevant to predictive performance, allowing retraining to be applied selectively rather than continuously.

In contrast, on the MIT-BIH ECG domain, entropy-triggered retraining substantially underperforms daily retraining despite a large reduction in retraining frequency. This behavior reflects a structural limitation of feature-space entropy monitoring in this setting: clinically relevant changes in ECG data often manifest as shifts in the conditional distribution $p(x \mid y)$ rather than in the marginal feature distribution $p(x)$. High intra-class variability driven by patient-specific morphology, noise, and recording conditions can dominate the KDE-based KL estimate, reducing its sensitivity to decision-relevant drift. As a result, entropy-triggered retraining may respond too weakly or too late when predictive performance degrades, even though substantial label-conditional drift is present.

For clarity of presentation, time-series plots are shown for the Wikipedia Pageviews domain as a representative example. This domain exhibits smooth, interpretable covariate drift and a clear performance–cost tradeoff, making it well suited for visualizing the behavior of entropy-triggered retraining. Analogous qualitative behavior is observed in the synthetic and financial domains, whose quantitative results relative to the maximum-frequency baseline are reported in Table 1.

| Domain | Daily Avg Loss | Entropy Avg Loss | Entropy Retrains (%) |
|---|---|---|---|
| Synthetic Gaussian Drift | 0.5587 | 0.5613 | 30 (15.0) |
| Finance (SPY) | 0.6984 | 0.7016 | 90 (2.0) |
| Wikipedia Pageviews | 0.6455 | 0.6377 | 68 (2.7) |
| Biomedical (MIT-BIH ECG) | 0.1293 | 0.3224 | 76 (1.5) |

Table 1: Cross-domain summary. "Daily" retrains at every step; percentages denote entropy-triggered retraining events relative to daily retraining within each domain.

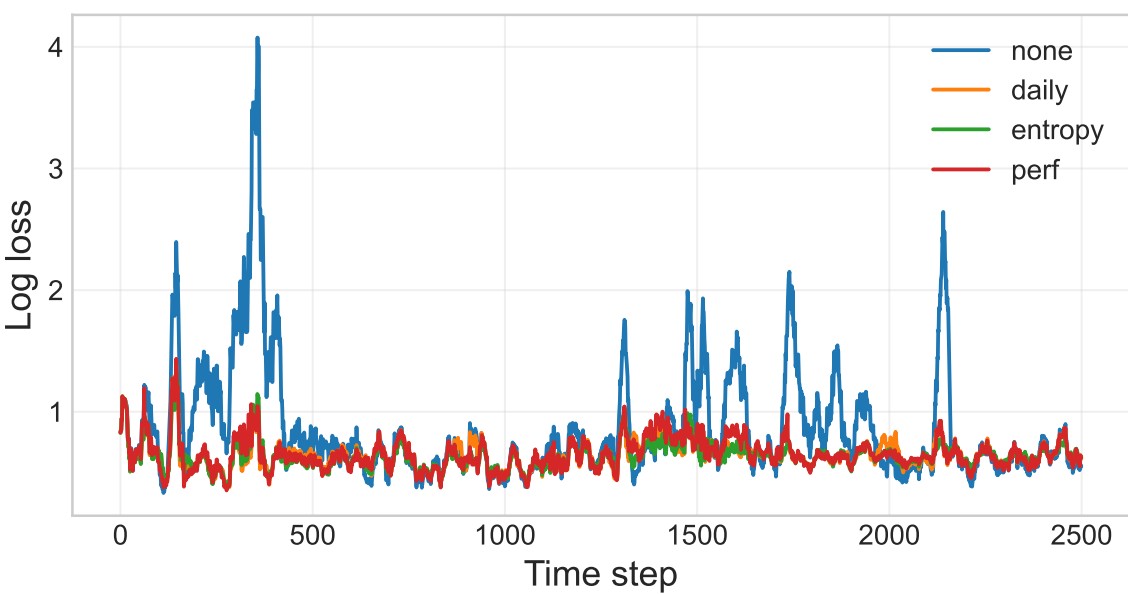

Figure 2: Predictive log loss over time under drift for the Wikipedia Pageviews domain.

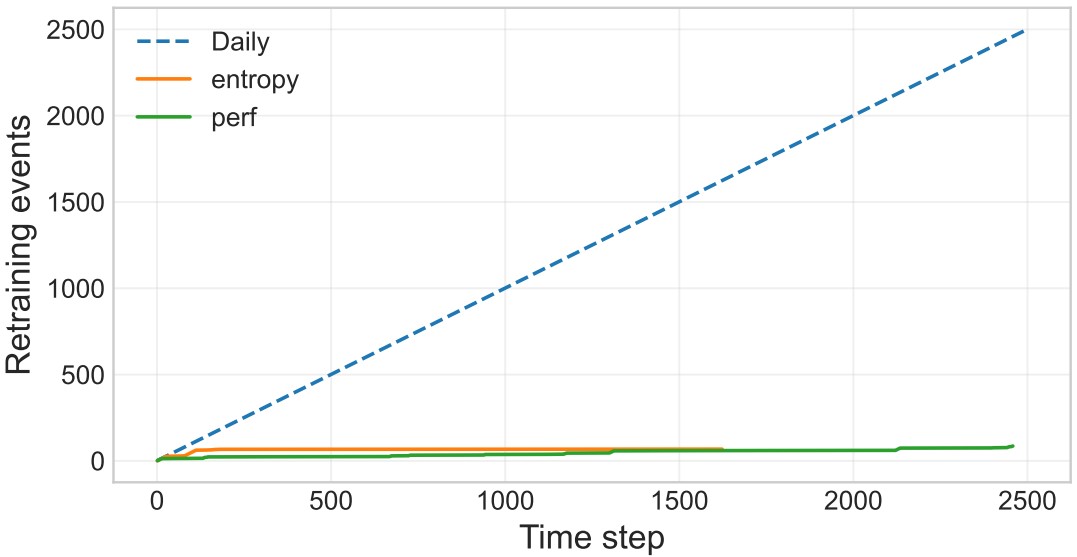

Figure 3: Cumulative number of retraining events over time for the Wikipedia Pageviews domain.

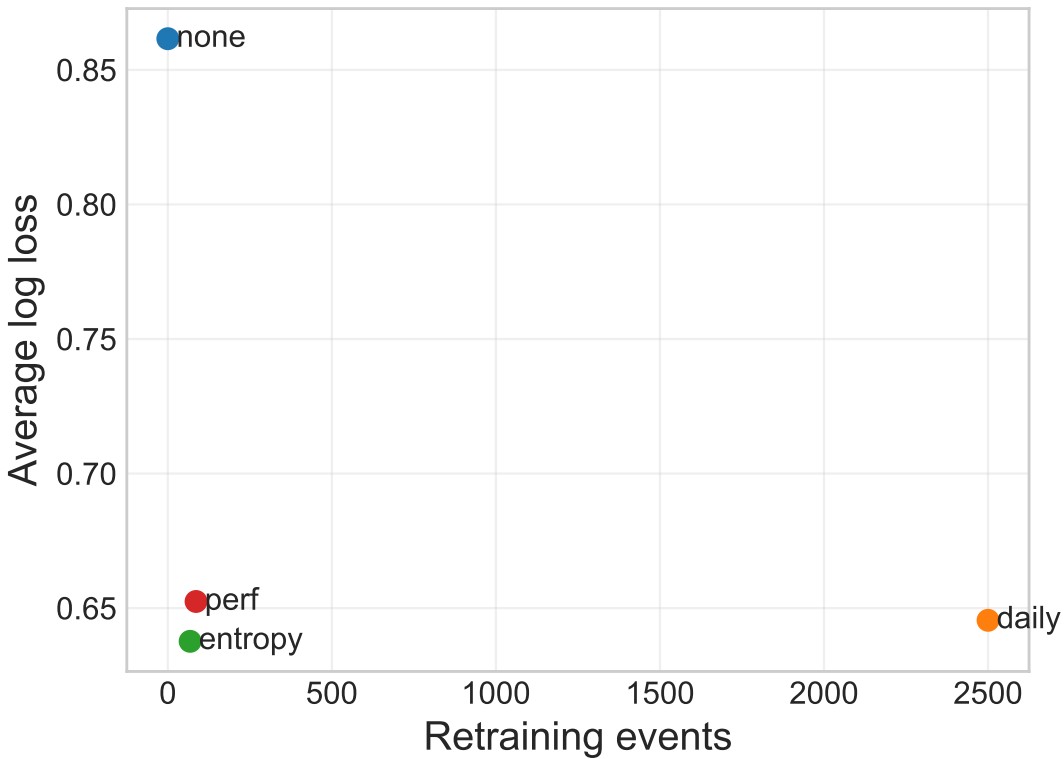

Figure 4: Cost–performance tradeoff (Pareto view) for the Wikipedia Pageviews domain.

## 6 Discussion

These results support the theoretical interpretation of model degradation as a nonequilibrium process driven by probability currents. Unlike heuristic drift metrics, mismatch entropy is tied directly to the dynamics of the deployment-time distribution and provides a principled handle for balancing predictive performance against retraining cost.

From a geometric viewpoint, relative entropy also admits an interpretation as a divergence on statistical manifolds, further supporting its role as a structural measure of model–data mismatch ichi Amari (2016). Practically, the main limitations concern estimation of mismatch entropy in high-dimensional settings and the choice of threshold. These are not unique to the proposed approach: any operational retraining policy requires selecting a tolerable performance–cost operating point. The advantage of entropy-triggered monitoring is that it connects this choice to an interpretable dynamical quantity driven by probability flow.

At the same time, the MIT-BIH ECG results indicate that a fixed, label-free drift signal and a fixed sensitivity $(h, k)$ can be insufficient for some domains. In such settings, more sensitive thresholds, domain-specific feature representations, or hybrid triggers combining supervised and unsupervised monitoring may be necessary.

## 7 Conclusion

We presented a theory-driven retraining framework that interprets model degradation under data drift as nonequilibrium entropy production. Modeling drift as Fokker–Planck probability flow yields a mismatch-entropy functional whose evolution is driven by probability currents and admits an entropy-balance decomposition with a nonnegative entropy production term. This provides a principled, label-free monitoring signal for deciding when retraining becomes necessary. Experiments demonstrate that entropy-triggered retraining with EWMA thresholding can achieve near-maximum-frequency baseline performance at a fraction of the retraining cost in several domains, but also reveal failure modes in which frequent retraining remains necessary.

## A Proof of the Entropy Production Decomposition

In this appendix we derive the decomposition of the time derivative of the model mismatch entropy that underlies the entropy-triggered retraining framework. The structure of the result parallels entropy-balance relations in stochastic thermodynamics for diffusion processes.

Let $p(x, t)$ be a probability density on $\mathbb{R}^d$ evolving according to the Fokker–Planck equation

$$\partial_t p(x, t) = -\nabla \cdot J(x, t), \tag{11}$$

with probability current

$$J(x, t) = a(x, t)p(x, t) - \nabla \cdot (D(x, t)p(x, t)), \tag{12}$$

where $a(x, t)$ is a drift field and $D(x, t)$ is a symmetric positive definite diffusion tensor almost everywhere.

Let $q_{\text{ref}}(x)$ be a fixed reference density satisfying $q_{\text{ref}}(x) > 0$ wherever $p(x, t) > 0$, and define the associated potential

$$U_\theta(x) = -\log q_{\text{ref}}(x). \tag{13}$$

We assume that:

- $p(\cdot, t)$ is smooth and integrable for each $t$,

- either $p(x, t) > 0$ almost everywhere or $J = 0$ on the set $\{p = 0\}$,

- boundary terms vanish so that integration by parts is valid.

The model mismatch entropy is defined as

$$D(t) = D_{\mathrm{KL}}\big(p(\cdot,t)\,\|\,q_{\mathrm{ref}}\big) = \int_{\mathbb{R}^d} p(x,t) \log \frac{p(x,t)}{q_{\mathrm{ref}}(x)}\,dx. \tag{14}$$

Differentiating with respect to time and using the continuity equation $\partial_t p = -\nabla \cdot J$, we obtain

$$\frac{d}{dt} D(t) = \int_{\mathbb{R}^d} \partial_t p(x,t) \log \frac{p(x,t)}{q_{\mathrm{ref}}(x)}\,dx \tag{15}$$

$$= -\int_{\mathbb{R}^d} \nabla \cdot J(x,t) \log \frac{p(x,t)}{q_{\mathrm{ref}}(x)}\,dx. \tag{16}$$

Integrating by parts and discarding boundary terms yields

$$\frac{d}{dt} D(t) = \int_{\mathbb{R}^d} J(x,t) \cdot \nabla \log \frac{p(x,t)}{q_{\mathrm{ref}}(x)}\,dx. \tag{17}$$

Using the definition of $U_\theta$, we write

$$\nabla \log \frac{p}{q_{\mathrm{ref}}} = \frac{\nabla p}{p} + \nabla U_\theta. \tag{18}$$

In the Itô formulation, the probability current satisfies

$$J = ap - \nabla \cdot (Dp) = \big[a - (\nabla \cdot D)\big]p - D\nabla p. \tag{19}$$

Solving for $\nabla p/p$, we obtain

$$\frac{\nabla p}{p} = D^{-1}\big[a - (\nabla \cdot D)\big] - \frac{D^{-1}J}{p}. \tag{20}$$

Substituting this expression into the entropy rate identity yields

$$\frac{d}{dt} D(t) = \int_{\mathbb{R}^d} J(x,t) \cdot \left(D^{-1}\big[a - (\nabla \cdot D)\big] + \nabla U_\theta\right) dx - \int_{\mathbb{R}^d} \frac{J(x,t)^\top D^{-1} J(x,t)}{p(x,t)}\,dx. \tag{21}$$

We define the total entropy production rate as

$$\dot{\Sigma}_{\mathrm{tot}}(t) := \int_{\mathbb{R}^d} \frac{J(x,t)^\top D(x,t)^{-1} J(x,t)}{p(x,t)}\,dx, \tag{22}$$

and group the remaining contribution into a housekeeping term $\dot{Q}_{\mathrm{hk}}(t)$.

Thus, the mismatch entropy satisfies the decomposition

$$\frac{d}{dt} D(t) = -\dot{\Sigma}_{\mathrm{tot}}(t) + \dot{Q}_{\mathrm{hk}}(t). \tag{23}$$

Since $D(x,t)$ is symmetric positive definite, its inverse $D^{-1}(x,t)$ is also positive definite. Consequently,

$$J(x,t)^\top D(x,t)^{-1} J(x,t) \geq 0 \tag{24}$$

pointwise, and division by $p(x,t) > 0$ preserves this inequality. Hence,

$$\dot{\Sigma}_{\mathrm{tot}}(t) \geq 0, \tag{25}$$

with equality if and only if $J(x,t) \equiv 0$, corresponding to equilibrium dynamics.

This establishes nonnegativity of the entropy production term in the mismatch-entropy balance relation used in the main text.

## B    EWMA Thresholding Details

This appendix records the EWMA trigger used for both entropy- and performance-triggered retraining in the experimental script.

Let $h$ be the half-life and define $\alpha = 1 - 2^{-1/h}$. Let $s_t$ be the monitored scalar statistic at time $t$ (either $\widehat{D}_t$ or an online log-loss estimate). The EWMA recursion is

$$\mu_t = (1 - \alpha)\mu_{t-1} + \alpha s_t, \tag{26}$$

$$v_t = (1 - \alpha)v_{t-1} + \alpha(s_t - \mu_t)^2. \tag{27}$$

We trigger retraining when $z_t > k$, where

$$z_t = \frac{s_t - \mu_t}{\sqrt{v_t} + \varepsilon}. \tag{28}$$

In experiments, $h = 50$, $k = 2.0$, and $\varepsilon = 10^{-12}$.

## Acknowledgements

**Reproducibility**   Computational experiments were conducted using a Jupyter notebook and a conda-managed Python environment. The notebook and environment specification can be shared upon request.

**Acknowledgement of Computational Resources**   The author gratefully acknowledges the use of a Dell Pro Max workstation for conducting the computational experiments reported in this work.

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
