# OpenReview forum: "Entropy-Triggered Retraining as Nonequilibrium Entropy Production in Deployed Machine Learning Systems"
_TMLR — Withdrawn by Authors_

### Review · Reviewer_kUkm · 2026-03-24

**Summary Of Contributions:**

### Summary
Machine learning models are typically trained under the assumption of independent and identically distributed (IID) data. However, this assumption does not hold in nonstationary environments, where conditions can change over time. Situations such as distribution shifts can lead to a decline in model performance because the model was trained on a different distribution. This work proposes a retraining strategy that addresses this mismatch using an entropy-based approach. Instead of retraining the model at each iteration (or daily), the suggested method involves retraining only when necessary. This decision is based on a threshold derived from an entropy measurement. The approach was evaluated across multiple data streams, demonstrating promising results compared to some baseline methods, while also highlighting certain limitations of the current approach.

### Strengths
- The work presents a well-founded proposal, showing the advantage of using an entropy-triggered retrain policy, instead of retraining after a fixed period of time.

### Weaknesses
- The paper discusses the need to update a model when a sequence distribution shift occurs; however, it overlooks a significant body of prior work relevant to this topic. Some examples of relevant areas that can be explored:
    - Work on distribution shifts for machine learning models across different types.
    - Detection of distribution shifts.
    - Work in the field of continuous learning. Although the vast majority are based on deep learning models, but these can easily be replicated in environments using smaller models.
    - These research areas share a similar motivation: in use in the paper: to minimise the need to retrain the model completely.
- The advantage of using the proposed method over periodic retraining is unclear.
    - Is there a gain in computational cost? When? Often, the cost of retraining these models is minimal, so retraining one every day would not be a problem.
    - Is there a gain in not having to label new data? How much? The amount of data used in each retraining is not specified; it would be interesting if the proposed method required less labelled data.

### Questions
- What distinguishes the framework used here from a traditional continual learning framework?
    - In continuous learning, there is a scenario known as 'domain incremental', where the classes from the first task are retained, but the distribution P(x) changes.
- It is well documented in previous work that there are various types of distribution changes, not just the two mentioned in the document. How does the proposed method behave in the face of these changes?
    - For example, what happens if the change in the distribution is more abrupt or slower?
    - What happens if it is periodic or temporary?

**Audience:**

Yes

**Audience Explanation:**

If the work were more exhaustive, it could attract interest from a part of the community. There is growing interest in understanding how models behave when they encounter distribution shifts and in exploring ways to adapt them over time. While this work is moving in that direction, several experiments still need to be conducted.

**Broader Impact Concerns:**

There are no concerns on the ethical implications.

**Claims And Evidence:**

No

**Claims Explanation:**

- The experiments confirm that the method works under certain conditions, but further experiments are needed to verify its performance across different contexts fully. These include different levels of distribution changes, different models, etc.
- A study of ablation is required at both the hyperparameter and model levels. The results focus only on a small number of variations.
- A more thorough comparison with previous work is needed.

**Requested Changes:**

- It would be helpful if the authors highlight the differences from methodologies and frameworks used in other research areas, such as continual learning or distribution shifts. This could be done by providing a clearer explanation and comparison, or by including some of these methods in the experiments.
- It would be helpful to include more baseline methods based on previous work.
- Given that synthetic datasets are used, it would be worth exploring how the methods behave under different distributional variations: more abrupt or slower changes, and temporal changes.
- It would be helpful to include ablation results for the hyperparameters used.
- A better explanation of the experimental setup used is lacking.
- The figures are neither used nor explained. It would be good to reference the figures in the text and explain the results they show.

---

> ### Author Response · Authors · 2026-03-24
> **Response to reviewer kUkm**
>
> Thanks for the your thoughtful feedback. We’re encouraged that you find the idea potentially interesting and the overall proposal well-founded. We agree the paper needs clearer positioning, a more explicit practical motivation, and a more exhaustive experimental study. We also agree the current draft doesn’t do enough to distinguish this setting from adjacent areas like continual learning, drift detection, and broader distribution-shift work. In the revision, we’ll make this distinction much clearer and add a more direct comparison to those frameworks.
>
> On the question of why not simply retrain every day, we agree that this needs to be more explicitly addressed in the manuscript. Daily retraining is meant as a high-frequency baseline, not necessarily as a practical recommendation. In many real settings, retraining is simply too expensive, on the order of hours or even days. In such cases, it’s important to minimize how often retraining is required while still maintaining good predictive performance. We’ll revise the paper to make this motivation much more explicit.
>
> We also agree the experimental section should be more exhaustive. In particular, we should include stronger comparisons to prior approaches, clearer discussion of the setup, and more ablations and sensitivity analyses. It would also strengthen the paper to test a broader range of drift regimes, including more abrupt, slower, and temporary changes. We’ll also improve how the figures are used and explained in the text. More broadly, we accept the point that the current experiments support the method under some conditions, but aren’t yet broad enough to fully establish its behavior across settings. We’ll revise accordingly and better clarify both the scope of the method and its limitations.
>
> We appreciate the suggestions and believe they’ll help us significantly strengthen the paper.

---

### Review · Reviewer_9uKK · 2026-03-25

**Summary Of Contributions:**

The authors start from the hypothesis that the pdf of the data is governed by a theoretical Fokker-Plank model. They then use this model to show that under this assumption the time derivative of the KL divergence of the data distribution w.r.t. a reference one is a function of the probability current induced by drift diffusion. Furthermore, they report a decomposition of the probability current introduced in the model.

The idea of retraining based on theoretical entropy production is fascinating, but experiments don't provide any evidence this lead to superior performances.

Overall the contributions seem limited.

**Audience:**

No

**Audience Explanation:**

For the reasons stated above, namely:
* the theoretical framework is not functional in conceptually supporting the proposed retraining strategy
* no evidence of superiority of the proposed strategy w.r.t. the proposed baseline

Additionally, the literature review is absent, and previous work in the field ignored. As model retraining and generalisation under data drifting and covariate shift are extensively explored/studied topics, a sound literature review is needed.

**Broader Impact Concerns:**

-

**Claims And Evidence:**

No

**Claims Explanation:**

* Assuming data evolves with the FP model seems a very specific assumption. It could represent a nice framework if
   * modelling assumptions are clearly correct / justified or if authors provide some insight on the type of data generation process for which the modelling assumption could hold
   * empirical results show superiority of the method.

   None of the above is provided in the paper.
* The decomposition of the probability current is not functional for the proposed strategy. As the authors state “In the present work, this decomposition serves as a theoretical motivation rather than a directly estimated quantity”. I disagree that this decomposition represent a theoretical motivation of the proposed strategy. In my view this would lead to an interpretation of the proposed strategy if the FP model was correct. The interpretation (not clearly provided in the paper) could be that retraining is done when drift dominates entropy production. A strategy based on e.g. the ratio of the two terms would lead to a clearer interpretation, but the decomposed terms can't be easily estimated.
* The decomposition per se seems to be a standard thermodynamic result, see e.g. "Stochastic thermodynamics, fluctuation
theorems and molecular machines, Udo Seifert" eq 31 or the reference from the same author in the manuscript.
* The experimental doesn't support superiority of the theoretical framework, in terms of better performances with respect to a baseline strategy based on loss monitoring. This could mean the modelling assumptions are not correct.
* In the experimental study, the proposed method should also be compared against dummy benchmarks, as random retraining under same retraining budget: average performance over n runs of training at m uniform random times, where m is the total number of retraining from the entropy-based strategy for that specific dataset (30, 90, 68, 76), or fixed-frequency retraining (as daily strategy) under same budget.

**Requested Changes:**

Major
* Add a proper literature review
* Add benchmarks/baselines (see comment above)
* Tune policy parameters or show sensitivity of results wrt chosen values
* quantify data distribution changes in the tested datasets

Minor

* Which distribution are you modelling in (2)? Is it the data distribution (property of environment) or the error/residual distribution (property of environment and model)? If the first (data distribution), why? Isn’t it enough to model error distribution for model retraining?
* I think D in (4) is not D in (3). If not, use another letter please
* You could mention that (5) is obtained by integrating by part, not directly from (2) and (4)
* “This identity makes explicit that the temporal evolution of mismatch is driven by probability currents in feature space induced by drift.” Just because you modelled it like this. I suggest a rephrasing.
* Remark 1 is cryptic. E.g. what’s an “appropriate stationary reference for the flow”? “D(t) need not be monotone” w.r.t. what? Time?
* “The mismatch entropy D(t) monitors changes in the marginal feature distribution p(x) and is therefore most sensitive to covariate drift” . Why not to monitor the joint or also the target distribution mismatch?
* From plot 1 seems to show a policy failure: since both numerator and denominator in (10) are defined by running statistics, an ever-increasing divergence estimation is likely to never trigger the retraining policy (step 150-500). But the plot indicates the data distribution significantly changed.
* In table 1 should be add perf baseline
* Would have been nice to plot also running averages of prediction error and KL divergence estimation of the target in figure 1
* Policy parameters should have been tuned (k, alpha)
* experiments: are there actually data drifts in the dataset selected? This should have been quantified.
* Covariate drifts doesn’t imply joint covariate/target drift. At least the target distribution shift should have been considered in the retraining policy.
* “For clarity of presentation, time-series plots are shown for the Wikipedia Pageviews domain as a representative example” in which picture?
* “These results support the theoretical interpretation of model degradation as a nonequilibrium process driven by probability currents.” I think this sentence is definitely too strong for the level of evidence provided.

---

> ### Author Response · Authors · 2026-03-25
> **Response to reviewer 9uKK**
>
> Thanks for your detailed feedback. We agree the current version needs clearer positioning, a more careful presentation of the theoretical framework, and a more exhaustive experimental study. We also strongly agree that the literature review is too sparse, and in the revision we’ll significantly expand it and more clearly position the work relative to prior research on drift detection, retraining strategies, and nonstationary learning.
>
> Regarding the Fokker-Planck perspective, we agree its role wasn’t clearly explained. Our intent is not to assume that data drift strictly follows an FP process, but to use it as a modeling tool. We also agree the manuscript doesn’t clearly separate theoretical motivation from what’s operational in the method; the decomposition is intended as a conceptual tool rather than a directly used quantity. We’ll revise the paper to clarify these points and avoid overstating the role of the theory.
>
> On the empirical side, we agree the current results aren’t yet sufficient to establish superiority. We’ll expand the experiments with stronger baselines, including budget-matched comparisons (e.g., random and fixed-frequency retraining under the same budget), along with additional ablations, sensitivity analyses, and a broader range of drift settings. More broadly, we agree some claims are currently too strong given the level of evidence, and we’ll revise them accordingly.

---

### Review · Reviewer_i4E3 · 2026-03-31

**Summary Of Contributions:**

The paper deals with an interesting problem of retraining machine learning models in deployment, considering the issue of distribution drift. The proposed solution is motivated by the principles of non-equilibrium thinking, which means that the solution is based on the Fokker-Planck probability flow, relative entropy, and a retraining condition based on relative entropy and an EWMA of the online KL divergence. The proposed solution was shown to be effective in various domains, with fewer retraining episodes, though it was significantly worse on the ECG dataset.

**Audience:**

No

**Audience Explanation:**

The general idea behind the paper seems interesting and promising. However, in its current form, it is hard to recommend it for publication. The paper is meant to tackle an important theoretical problem. However, in its current form, it is not entirely clear that the technical development is adequate for the claims that the paper makes. Specifically, there seems to be a significant gap between what is being claimed in the paper and what is actually proved in the manuscript. Therefore, I would strongly suggest that the authors revisit their work, think carefully about what they have accomplished, and then resubmit once all of these issues have been addressed.

**Claims And Evidence:**

No

**Claims Explanation:**

The biggest weakness of the paper, in my humble opinion, is that there is a type of presentation inflation throughout the paper, where the framing implies that there is a much more fundamental and tight connection between the proposed method of retraining and nonequilibrium statistical physics than the paper actually supports. Early in the paper, there is significant emphasis given to concepts such as entropy production, probability currents, and continuous-time diffusion dynamics, which implies that the proposed method is derived from or at least rigorously justified through nonequilibrium statistical physics. However, later in the paper, we learn that the SDE/Fokker-Planck perspective is merely a conceptual interpretation and is not actually assumed to apply to the underlying data streams. This is actually a crucial clarification, but unfortunately, it comes too late in the paper, after the more forceful framing has already created the reader’s expectations. This creates problems with the theory-method connection because the actual contribution of the paper is, in the end, much more limited and conventional than the paper initially implies: the procedure that is implemented is simply a drift monitoring heuristic using an EWMA alarm triggered on the basis of the computed KL divergence using the KDE.

There is nothing inherently wrong with this contribution, but the paper simply does not do enough to demonstrate that the connection to nonequilibrium physics leads to a better algorithm, sharper theory, or more novel retraining principle beyond the narrative level. The same is true of the theoretical sections: the KL derivative identity along the continuity equation and the entropy balance equation with nonnegative production term are discussed with considerable conceptual emphasis, but the paper does not provide estimates of the relevant terms in practice, nor does it derive from them a concrete decision rule that is actually employed in the experiments. This would be understandable if the paper were simply more forthcoming about the fact that these derivations are more interpretive than concrete, but the paper appears to waver between the stance of a theory paper and the stance of an applied monitoring paper. This is confusing to the reader because the language and framing are more appropriate to the former, but the actual content and validation are more appropriate to the latter. A clearer paper would be more explicit about its claims at the outset, would clearly distinguish between conceptual and algorithmic content, and would make more limited but more defensible claims, such as the paper is proposing a label-free drift monitoring heuristic for retraining, motivated from the perspective of entropy, which is useful in the context of covariate drift but which has yet to establish the more general nonequilibrium perspective that the paper appears to be gesturing at.

**Requested Changes:**

While I think the paper would need substantial revision before it would be suitable for publication, my main worry is not that the underlying idea is entirely uninteresting, but that the paper does not yet sufficiently clearly, rigorously, and comprehensively present the underlying idea. As the paper is now, there is some uncertainty about whether the paper is intended to be read as a theory paper or an applied monitoring paper. While the paper is somewhat repetitive, the underlying high-level intuition is constantly repeated using slightly different wording but without adding real substance. As a consequence, the theoretical contribution is not well defined. One of the main problems is the paper's constant use of nonequilibrium statistical physics, entropy production, and related continuous-time diffusion concepts. However, the paper leaves unclear how much of this is intended to be taken rigorously as part of the methodological contribution and how much is intended to be taken more interpretively.

 As a consequence, there is the feeling of "presentation inflation": the paper's rhetoric implies that the connection between theory and method is more general and fundamental than the paper actually delivers. Indeed, the actual contribution of the paper is, at the level of detail, relatively limited and conventional: the contribution is essentially the combination of the KDE-based KL-divergence estimation and the EWMA-based alarm, but the paper does not sufficiently clearly establish how the "nonequilibrium perspective" contributes to this choice of statistic. As the paper is now, the paper would be greatly improved if the underlying idea were more clearly and sharply presented.

For the next major revision, the following points would be good to keep in check:
1. What claims are the paper making about its methodological contribution, and which are simply interpretive? Specifically, is the Fokker-Planck/entropy production machinery intended to be taken literally as a modeling assumption, or is it more of an analogy? And is the machinery intended to be taken literally as providing new algorithmic structure?

2. What is the concrete benefit of the nonequilibrium perspective beyond providing the KL-based drift statistic? Put another way, what are we to gain from the paper beyond the sum of its parts: KDE-based KL estimation and EWMA thresholding?

3. How robust are the findings are to the choice of the KDE bandwidth, the EWMA half-life h, and the threshold k? The paper does not make it entirely clear whether the performance trade-offs are robust or required significant parameter tuning.

4. As the paper itself points out, label conditional drift is a critical failure case. How is the reader to know in advance whether the trigger is appropriate in their case?

---

> ### Author Response · Authors · 2026-03-31
> **Response to Reviewer i4E3**
>
> Thank you for the detailed review. We agree with your main concern that the paper doesn’t clearly separate conceptual framing from the actual method, which creates a mismatch between presentation and implementation. We also concede your point about how the nonequilibrium statistical physics perspective is introduced. Our intent isn’t to assume a Fokker–Planck model for the data, but to use it as a way to interpret drift and motivate relative entropy as a monitoring signal. While this is stated in the paper, we acknowledge that it appears too late and doesn’t sufficiently qualify the earlier framing. In the revision, we’ll make this explicit at the outset and adjust the language to avoid overstating the theoretical role.
>
> We agree that your concern about the theory–method connection is valid. The entropy-based derivations are meant to motivate the use of KL divergence, but they aren’t directly used in the retraining rule. As you point out, the implemented method is a KL-based statistic with EWMA thresholding, and we’ll state this more directly. At the same time, we would position the theoretical contribution as providing structure for why this statistic is reasonable, rather than as a derivation of the full procedure. We also agree that your point on positioning is important. In the revision, we’ll frame the work more precisely as a label-free drift monitoring and retraining heuristic with a dynamical interpretation, and we’ll moderate claims to better match the level of theoretical and empirical support.
>
> On the empirical side, we agree with your concerns about robustness and concede that the current draft doesn’t sufficiently analyze sensitivity to KDE bandwidth, EWMA parameters, or thresholds. We’ll add these results. We also agree that your point about label-conditional drift is important. While discussed, it should be more central, and we’ll clarify when the method is expected to work, when it may fail, and how to assess applicability. Overall, we agree that the main issue is clarity and positioning, and we’ll revise the paper accordingly.

---

### Note · Authors · 2026-05-10

**Comment:**

Withdrawn for substantial revisions

**Withdrawal Confirmation:**

I have read and agree with the venue's withdrawal policy on behalf of myself and my co-authors.